# Resveratrol Alleviates Advanced Glycation End-Products-Related Renal Dysfunction in D-Galactose-Induced Aging Mice

**DOI:** 10.3390/metabo13050655

**Published:** 2023-05-13

**Authors:** Kuo-Cheng Lan, Pei-Jin Peng, Ting-Yu Chang, Shing-Hwa Liu

**Affiliations:** 1Department of Emergency Medicine, Tri-Service General Hospital, National Defense Medical Center, Taipei 114202, Taiwan; kclan.tw@mail.ndmctsgh.edu.tw; 2Graduate Institute of Toxicology, College of Medicine, National Taiwan University, Taipei 100233, Taiwan; r04447004@ntu.edu.tw (P.-J.P.); d03447003@ntu.edu.tw (T.-Y.C.); 3Department of Medical Research, China Medical University Hospital, China Medical University, Taichung 404333, Taiwan; 4Department of Pediatrics, College of Medicine, National Taiwan University & Hospital, Taipei 100233, Taiwan

**Keywords:** aging, advanced glycation end-products, resveratrol, renal dysfunction, renal fibrosis

## Abstract

The elderly have higher concentrations of advanced glycation end-products (AGEs). AGEs are considered risk factors that accelerate aging and cause diabetic nephropathy. The effects of AGEs on renal function in the elderly remain to be clarified. This study aimed to explore the role of AGEs in renal function decline in the elderly and the protective effect of resveratrol, a stilbenoid polyphenol, comparing it with aminoguanidine (an AGEs inhibitor). A D-galactose-induced aging mouse model was used to explore the role of AGEs in the process of renal aging. The mice were administered D-galactose subcutaneously for eight weeks in the presence or absence of orally administered aminoguanidine or resveratrol. The results showed that the serum levels of AGEs and renal function markers BUN, creatinine, and cystatin C in the mice significantly increased after the administration of D-galactose, and this outcome could be significantly reversed by treatment with aminoguanidine or resveratrol. The protein expression levels for apoptosis, fibrosis, and aging-related indicators in the kidneys were significantly increased, which could also be reversed by treatment with aminoguanidine or resveratrol. These findings suggest that resveratrol could alleviate AGEs-related renal dysfunction through the improvement of renal cellular senescence, apoptosis, and fibrosis in D-galactose-induced aging in mice.

## 1. Introduction

Due to the aging global population, geriatric concerns have been gradually receiving more attention. The great improvement in human lifespan over the past decade has caused a dramatic increase in the elderly population worldwide. According to the World Health Organization (WHO), the number of people over 60 years of age is projected to grow from an estimated 1 billion in 2020 to 1.4 billion in 2030, and it will double, reaching to 2.1 billion, by 2050 [1]. According to the survey results of the US Centers for Disease Control and Prevention, chronic kidney disease (CKD) is more prevalent in US adults aged 65 years or older (38.1%) than those aged 45–64 years (12.4%) and 18–44 years (6%) [2]. In the aging process, the kidneys undergo structural and functional changes [3,4]. A previous study observed that smaller kidney volumes were correlated with more advanced age among more than 600 adult volunteers [5]. Some studies have also indicated that kidney volume starts to decrease after 50 years of age [6,7]. The predominant molecular changes in aging kidneys concern cellular senescence, which is the state when cells undergo permanent and irreversible growth arrest [8]. Senescent cells present altered morphologies, expression levels of senescence-associated β-galactosidase, greater heterogeneity, and the accumulation of lipofuscin granules [9].

Advanced glycation end-products (AGEs) are complexes produced by non-enzymatic reactions between reduced sugars and free amino acids and nucleic acids, and the human body generates AGEs during normal metabolism [10]. The accumulation of AGEs occurs during the aging process and in those with diabetes mellitus [11,12], and AGEs are considered as risk factors that accelerate aging and cause diabetes nephropathy [10,11,12]. The toxic effects of AGEs are related to the induction of oxidative stress and inflammation by binding to cell surface receptors, such as the specific receptor for AGEs (RAGE), cross-linking with proteins, or altering their structure and function [13]. AGEs have attracted attention in diabetes nephropathy, and several mechanisms have been suggested to cause renal injury [11]. However, there have been few studies focused on the effects of higher AGEs on the kidneys of older individuals, in comparison to younger, healthier populations [14]. The relationship between AGEs and declining renal function during the aging process remains unclear.

The D-galactose-induced aging animal model is widely applied in anti-aging research [15,16,17,18]. Several natural extracts and agents, such as chlorogenic acid in green coffee beans, have been shown to have anti-aging activities using this animal model [15]. Most of these natural anti-aging extracts have anti-oxidative properties. Song et al. reported that D-galactose-treated C57BL/6 mice had significantly increased levels of AGEs in serum [16]. After aminoguanidine (an AGEs inhibitor) treatment, the AGEs production was effectively prevented and the neurological impairment and decreased activity due to anti-oxidant enzymes, such as superoxide dismutase, were prevented [16]. These researchers also demonstrated that D-galactose- or AGEs-treated mice had symptoms that resembled those of mice that had aged naturally [16]. Resveratrol, a stilbenoid polyphenol, has been considered as not only an anti-oxidant but also an anti-aging agent [19]. Resveratrol attenuated renal injury and fibrosis by inhibiting the epithelial-to-mesenchymal transition (EMT) [20]. Interestingly, resveratrol was able to decrease the levels of AGEs in serum and protect renal function in streptozotocin-induced diabetic rats by ameliorating hyperglycemia-mediated oxidative stress and inflammatory cytokines [21]. However, the mechanism of action of aminoguanidine or resveratrol on aging kidneys remains to be clarified.

In this study, we hypothesized that resveratrol had the potential to alleviate the AGEs-related renal dysfunction through cellular senescence, EMT, and apoptosis in aging individuals. The aim of this study was to investigate the effect of resveratrol on AGEs-related renal dysfunction, and compare it with aminoguanidine, in a D-galactose-induced aging mouse model, which included the resultant oxidant damage due to aging [17].

## 2. Materials and Methods

### 2.1. Animals

Male C57BL/6 mice at 10 weeks old were obtained from the Animal Center at the College of Medicine, National Taiwan University (Taipei, Taiwan). All the mice were housed at room temperature with 12 h light/night cycles and supplied with unlimited water and food. This animal study was approved by the Institutional Animal Care and Use Committee at the College of Medicine, National Taiwan University, and it was conducted in accordance with the guidelines for the care and use of laboratory animals.

The mice were randomly divided into four groups. After the acclimation period, the mice were given one of the following daily treatments for 8 weeks: (1) normal control group; (2) D-galactose treatment group; (3) D-galactose + aminoguanidine (100 mg/kg body weight (bw); or (4) D-galactose + resveratrol (40 mg/kg bw). The animal number per group was eight. D-galactose (Sigma-Aldrich, Louis, MO, USA) dissolved in PBS was injected subcutaneously to the mice at 1 g/kg bw. Aminoguanidine and resveratrol (Sigma–Aldrich) were dissolved in deionized water and 0.5% carboxymethyl cellulose sodium salt (Sigma–Aldrich), respectively, and both were administered orally by gavage to mice [18,22]. The mice in the D-galactose group were also orally administered with carboxymethyl cellulose solution (vehicle). The mice in the normal control group were subcutaneously injected with PBS (vehicle) and orally administered with carboxymethyl cellulose solution (vehicle). The dosage selection for aminoguanidine and resveratrol was determined according to previous studies [21,22] and our preliminary experiments. The body weights of all the mice were measured weekly. Mice were euthanized at the end of the experiment. Samples of blood were collected and centrifuged (3000 rpm, 15 min) to obtain the sera. The kidneys from all mice were isolated for immunoblotting detection and histological examination.

### 2.2. Biochemical Measurement

The serum levels of blood urea nitrogen (BUN), creatinine, and cystatin C were detected to evaluate renal function. A clinical chemistry analyzer (Roche, Rotkreuz, Switzerland) was used to detect the levels of serum creatinine and BUN. Cystatin C levels were measured in each mouse using a competitive ELISA kit (catalog #: E-90CYS; ICL Lab, Portland, OR, USA), which the absorbance at 450 nm was detected. Serum AGEs levels were also determined by competitive ELISA assay (catalog #: STA-817; Cell Biolabs, San Diego, CA, USA), during which the absorbance at 450 nm was detected.

### 2.3. Histological Examination

Kidney tissues were harvested, fixed with 10% buffered formalin (formaldehyde 3–5% and methanol 1–3%), embedded in paraffin, and cut into 4 μm thick sections. Periodic acid–Schiff (PAS) staining was used to observe renal histological injury. Tissue sections were deparaffinized in xylene and rehydrated through a graded series of alcohols to water. Tissue sections were incubated with 0.5% periodic acid solution for 5 min, stained with Schiff reagent for 15 min, then counterstained with hematoxylin solution for 1–2 min. Tissue sections were dehydrated and mounted with a coverslip. Histological score was divided into 4 grades, from 0 to 4 (0, no changes; 1, sample changes affecting <25%; 2, sample changes affecting 25–50%; 3, sample changes affecting 50–75%; and 4, sample changes affecting 75–100%) based on mesangial expansion, tubular atrophy, cast formation, an increase in the thickness of basement membranes and interstitial area, and clusters of inflammatory cells. Tissues sections were stained with Masson’s trichrome for detection of fibrosis. The fibrotic fraction was calculated using Image J software. For histological examination, fifteen fields per section per animal were assessed.

### 2.4. Immunoblotting Analysis

For immunoblotting, kidney cortical lysates were prepared. Kidney sections (10–15 mg) were lysed in 200 μL radioimmunoprecipitation assay (RIPA) lysis buffer at a pH of 7.4 (20 mM Tris-base, 150 mM NaCl, 1 mM EDTA, 1 mM EGTA, 1% NP40) and supplemented with a protease and phosphatase inhibitor cocktail. Lysates were centrifuged at 13,000 rpm, 4 °C for 30 min. Then, the supernatants were collected, and protein concentration was quantitated by bicinchoninic acid (BCA) assay (Thermo Scientific, IL, USA). Samples were boiled for 5 min with 4X sample buffer (8% SDS, 20% 2-ME, 40% glycerol, 0.008% bromophenol blue, 250 mM Tris-HCl), and separated by 8–15% sodium-dodecyl sulfate-polyacrylamide gel electrophoresis (SDS-PAGE). Separated proteins were transferred onto 0.45 µm polyvinyl-difluoride (PVDF) membranes (Merck Millipore, MA, USA) using a wet transfer tank filled with Tris-glycine buffer (25 mM Tris, 192 mM glycine, pH 8.3) containing 20% methanol. The membranes were blocked with 5% fat-free milk in TBST (20 mM Tris-base, 0.137 M NaCl, 0.2% Tween-20, pH 7.4) at room temperature for at least 1 h. Then, the membranes were incubated with the primary antibodies: anti-β-actin (sc-47778; 1:1000), anti-p21 (sc-397; 1:1000), anti-CCAAT/enhancer-binding protein (C/EBP) homologous protein (CHOP) (sc-575; 1:1000), anti-p53 (sc-126; 1:1000) (Santa Cruz, CA, USA), anti-α-SMA (A2547; 1:1000 ; Sigma Chemical, MO, USA), anti-Caspase-3 (#9662; 1:1000), anti-connective tissue growth factor (CTGF; #86641; 1:1000), and anti-Bax (#2772; 1:1000) (Cell signaling, Danvers, MA, USA) overnight at 4 °C. The next day, the membranes were washed by TBST 3 times (10 min each time) and then incubated with horseradish peroxidase (HRP)-conjugated secondary antibodies for 1 h at room temperature. Subsequently, the membranes were visualized with enhanced chemiluminescence (ECL) reagent (BioRad, CA, USA) and exposed to film. To quantify the intensity of the protein expression levels, densitometric analysis was assessed using Image J software.

### 2.5. Statistics

Data are presented as means ± SEM. A one-way analysis of variance, followed by Dunnett’s post hoc test, was used to assess the significant differences between each experimental sample and its respective controls. A *p*-value less than 0.05 indicated that the difference was statistically significant. GraphPad Prism software, version 6.0, was used to perform the statistical analysis.

## 3. Results

### 3.1. The Role of AGEs in Renal Dysfunction and the Protective Effect of Resveratrol in D-Galactose-Induced Aging Mice

We first investigated the role of AGEs in renal dysfunction in aging mice. The male C57BL/6 mice were subcutaneously injected with D-galactose at 1 g/kg bw/day for 8 weeks in the presence or absence of aminoguanidine (100 mg/kg bw) or resveratrol (40 mg/kg bw). As shown in Figure 1A,B, there were no statistically significant differences in the body weights and the relative kidney wet-weights among the four groups of D-galactose-injected mice over 8 weeks (*p* > 0.05). However, the serum AGEs levels significantly increased in the aging mice, as compared to the control mice, which could then be significantly reduced by treatments of either aminoguanidine or resveratrol (Figure 1C; *p* < 0.05). We next investigated the renal function in aging mice. As shown in Figure 2, the serum levels of the renal function markers BUN, creatinine, and cystatin C significantly increased in the aging mice, as compared to control mice, which could then also be significantly reversed by treatments of either aminoguanidine or resveratrol (*p* < 0.05). Moreover, the changes noted in the histological examination of the kidneys of the aging mice by PAS staining were observed, and the histological score was calculated. As shown in Figure 3, the D-galactose-induced aging mice showed aging features in the kidneys, such as glomerular mesangial expansion, basement membrane thickness, interstitial fibrosis, and the atrophy of renal tubules. The histological scores of the kidneys of the aging mice were significantly higher than those calculated for the control mice (*p* < 0.05). We further examined and quantified the changes in the fibrosis in the kidneys of aging mice by Masson’s trichrome staining. As shown in Figure 4, the moderate fibrosis was observed in the kidneys of aging mice. The fibrotic area in the kidneys of the aging mice was significantly higher than in those of the control mice (*p* < 0.05). The changes in the histological examination and the fibrosis analysis in the kidneys of the aging mice could be significantly reversed by treatments of either aminoguanidine or resveratrol (Figure 3 and Figure 4; *p* < 0.05). These results indicated that aminoguanidine or resveratrol could effectively reduce the AGEs formation and renal dysfunction in the D-galactose-induced aging mice.

### 3.2. The Changes in Signaling Molecules for AGEs-Related Kidney Injury and the Protective Effect of Resveratrol in D-Galactose-Induced Aging Mice

To examine the possible mechanisms of AGEs-related renal injury, we next investigated the changes in the signaling molecules for cellular senescence, fibrosis, and apoptosis in the kidneys of the D-galactose-induced aging mice.

We first tested the changes in the cellular-senescence-related signaling molecules by Western blot analysis. As shown in Figure 5A, the levels of protein expression for the cellular senescence markers of p21 and p53 significantly increased in the kidneys of the aging mice, as compared to the control mice (*p* < 0.05). The protein expression levels of CHOP, a signaling molecule for endoplasmic reticulum (ER) stress-related cellular senescence, also increased in the kidneys of aging mice compared to the control mice (Figure 5B; *p* < 0.05). Treatments with either aminoguanidine or resveratrol could significantly reverse the increases in p21, p53, and CHOP protein expression levels (Figure 5; *p* < 0.05).

We further examined the changes in the fibrosis- and apoptosis-related signals in the kidneys of D-galactose-induced aging mice. The epithelial-to-mesenchymal transition (EMT)-related fibrotic signaling molecules were first tested. As shown in Figure 6, the protein expression levels of α-SMA and CTGF significantly increased and the E-cadherin protein expression level significantly decreased in the kidneys of the aging mice, which could be significantly reversed by either aminoguanidine or resveratrol treatments (*p* < 0.05). We further found that the apoptosis-related signals in the kidneys of the aging mice were induced. As shown in Figure 7A, the levels of protein expression for the cleaved form of caspase-3 and Bax significantly increased in the kidneys of the aging mice (*p* < 0.05). The aminoguanidine or resveratrol treatments could effectively reduce the increases in the cleaved form of the levels of caspase-3 and Bax protein expression (Figure 7A; *p* < 0.05). In addition, the protein expression of α-Klotho, a key modulator of aging, significantly decreased in the kidneys of the aging mice, which could be significantly reversed by either aminoguanidine or resveratrol treatments (Figure 7B; *p* < 0.05).

## 4. Discussion

D-galactose is known for reducing the sugar that has the potential to form AGEs in vivo [16,17]. An aging mouse model induced by D-galactose was developed previously [17,23]. The increased AGEs levels may promote the process of aging, which has been shown in D-galactose-induced aging mice [16]. Overall, long-term D-galactose administration in mice has been suggested as a useful model to study insulin resistance, metabolic syndrome, and aging [24]. In this study, we further investigated the role of AGEs in renal dysfunction in D-galactose-induced aging mice, and explored the protective effect and mechanism of resveratrol compared with aminoguanidine, which has been shown to be an AGE inhibitor [25].

Cellular senescence is a critical element that has been closely associated with aging-related renal diseases [26,27]. When telomeres shorten and become inactive, cells interpret this as a break in the DNA strand and respond by activating the p53/p21^CIP1/WAF1^ pathway, causing cell-cycle arrest and inducing replicative senescence [26,28]. AGEs can induce cellular senescence in endothelial cells [29] and atrial myocytes [30]. Resveratrol has been shown to prevent high-glucose-induced cellular senescence in cultured human diploid fibroblasts [31]. CHOP signaling has been demonstrated to contribute to an ER-stress-activated alveolar epithelial cellular senescence, inducing pulmonary fibrosis [32]. In the present study, we found that the increased p53/p21 and CHOP protein expression in the kidneys of the aging mice could be reversed by treatments with aminoguanidine or resveratrol, suggesting that resveratrol has the potential to reduce the AGEs-related renal cell senescence during the aging process.

Shin et al. (2015) found that EMT and apoptosis could be induced by ER-stress inducers in human peritoneal mesothelial cells, suggesting that ER-stress-regulated EMT and apoptosis may contribute to peritoneal fibrosis in peritoneal dialysis [33]. An increase in CTGF expression in the epithelial cells of the skin and lung was found to trigger EMT-like morphological changes and α-SMA expression [34]. AGEs were shown to induce renal dysfunction that could have been elicited by an EMT-related podocyte injury and depletion [35]. Resveratrol has been considered to have both anti-oxidant and anti-aging properties. It was shown to have beneficial effects on the health and the survival of mice fed a high-calorie diet [19]. Resveratrol was shown to improve renal injury and fibrosis by suppressing the EMT process [20]. Guo et al. (2020) suggested that AGEs were a major stress in renal-accelerated aging in diabetes, and the anti-aging protein Klotho may play a fundamental role in cellular senescence in diabetic kidneys [36]. In the present study, we found that the increased expression of the EMT and apoptosis markers and the decreased α-Klotho protein expression in the kidneys of the D-galactose-induced aging mice could be significantly reversed by treatments with aminoguanidine or resveratrol, suggesting that resveratrol was capable of improving AGEs-related EMT-induced fibrosis and decreasing the protein Klotho in the kidneys.

There are several review articles discussing the effects of AGEs or resveratrol on aging. AGEs induce pathophysiological changes during the aging process [37,38]. AGEs are known aging products, and they participate in the onset and progress of chronic kidney disease [39]. Resveratrol has been suggested to have the potential to suppress the AGEs-related complications [40]. Resveratrol possesses anti-aging effects in the kidneys by decreasing pathological changes and regulating molecular signals, such as increased AMP-activated protein kinase (AMPK), decreased nuclear factor-κB, and increased sirtuin-1 [41]. Vlassara et al. (1994) observed the pathogenic role of AGEs in the induction of glomerular sclerosis and albuminuria in healthy rats administered intravenously with AGE-modified rat albumin, which could be ameliorated by aminoguanidine treatment [42]. Moreover, Abharzanjani and Hemmati (2021) found that resveratrol (10 and 15 mg/kg, i.p. for 3 weeks) could prevent kidney complications and aging in streptozotocin-induced diabetic rats [43]. Resveratrol (5 mg/kg, orally for 4 weeks) was found to decrease elevated serum AGEs levels and alleviate renal dysfunction in streptozotocin–nicotinamide-induced diabetic rats [21]. In [44], Grujić-Milanović et al. (2022) found that resveratrol (10 mg/kg, orally for 4 weeks) had a therapeutic potential in malignantly hypertensive rats against kidney injury [44]. However, the mechanisms of aminoguanidine or resveratrol on aging kidneys remains to be clarified. Compared with previous studies, this study has different approaches and novel findings. We investigated the protective effect and mechanism of resveratrol (40 mg/kg, orally for 8 weeks) against kidney aging and injury using a D-galactose-induced aging mouse model. We also compared the effects of resveratrol and AGEs inhibitor aminoguanidine to clarify the role of AGEs therewithin. We found that resveratrol could alleviate AGEs-related renal dysfunction by the decreases in renal cellular senescence, apoptosis, and fibrosis-related EMT in aging mice.

Several studies have employed experimental designs using the D-galactose-induced aging mouse model. Chu et al. (2021) showed that the expression levels of the Klotho gene and protein in the kidneys of D-galactose-injected (1 g/kg, intraperitoneal injection) mice were significantly lower than those of the control group, and the Klotho expression was enhanced by a resveratrol (12.5, 25, and 50 mg/kg, oral administration for 4 weeks) treatment [45]. Fan et al. (2016) observed the increased levels of AGEs in the kidneys of D-galactose-injected (120 mg/kg, subcutaneous injection) mice, as compared to a control group [15]. Li et al. (2021) found that resveratrol (10–40 mg/kg, oral administration for 8 weeks) significantly downregulated the abnormal increases in BUN and creatinine serum levels, caused by D-galactose (100 mg/kg, intraperitoneal injection) treatment, and effectively decreased pathological damage [46]. In the present study, we observed the increased serum levels of AGEs and the markers of renal function and renal histopathological damage and fibrosis, as well as the inductions of cellular senescence, fibrosis, and apoptosis in the kidneys of D-galactose-induced aging mice (1 g/kg, subcutaneous injection), and these changes could be significantly reversed by treatment with AGEs inhibitor aminoguanidine (100 mg/kg, orally) or resveratrol (40 mg/kg, orally) for 8 weeks. Using the D-galactose-induced aging mouse model, our study had a similar experimental design to the aforementioned studies, but we also had novel findings for simultaneously comparing the effects of aminoguanidine and resveratrol, including the inhibition of pathological characterizations determined by the histological score and collagen deposition (Masson’s trichrome staining) and the protective mechanisms of aminoguanidine or resveratrol against cellular senescence, fibrosis-related EMT, and apoptosis in aging kidneys. These findings suggest that AGEs play an important role in renal dysfunction during the aging process, and resveratrol treatment alleviates AGEs-related renal dysfunction through decreases in renal cellular senescence, apoptosis, and fibrosis-related EMT in D-galactose-induced aging in mice.

Sha et al. (2021) found that maltol reversed the increase in the protein expression of p53, p21, Bax, and caspase-3, induced by D-galactose treatment, and suggested that a phosphatidylinositol-3-kinase (PI3K)/protein kinase B (Akt) signaling pathway contributed to the protective effect of maltol against cellular senescence and injury of the liver and kidney in D-galactose-induced aging mice [47]. In the present study, we also found that resveratrol could effectively reverse the increased cellular senescence and apoptosis in the kidneys of D-galactose-induced aging mice. The PI3K/Akt signaling pathway may be involved in the protective effect of resveratrol against kidney aging and injury, which should be studied further in the future.

Resveratrol has been suggested to be a double-edged sword for its beneficial effects on health [48]. It exhibited several bioactive effects, such as anti-inflammatory, anticancer, cardiovasculoprotective, and neuroprotective effects. However, it presents some concerns for pharmaceutical industry, such as poor solubility and bioavailability and adverse effects [48]. The side effects of mild to moderate gastrointestinal symptoms by resveratrol have been observed at doses of 2.5 and 5 g per day for 29 days in healthy volunteers [49]. The dosage and route of administration involved in the safety of resveratrol still need to be considered and explored.

## 5. Conclusions

In this study, we used a D-galactose-induced aging mouse model to demonstrate the important role of that AGEs play in renal dysfunction in aging kidneys. In addition, we found that the increased serum AGEs levels and renal dysfunction, including cellular senescence, fibrosis, and apoptosis, in aging kidneys could be significantly reversed through treatment with aminoguanidine, a known AGEs inhibitor, or resveratrol, a potential therapeutic candidate. These findings also provided a therapeutic and/or preventive strategy against aging-related renal dysfunction. However, the detailed mechanisms for AGEs-related renal dysfunction/injury in the kidneys during the aging process still require further investigation.

## Figures and Tables

**Figure 1 metabolites-13-00655-f001:**
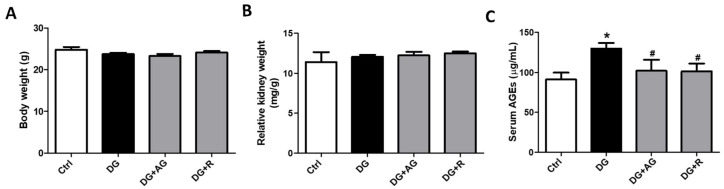
The bodyweight, kidney wet-weight, and serum AGEs level in D-galactose-induced aging mice. Mice were treated with D-galactose (DG) for 8 weeks in the presence or absence of either aminoguanidine (AG) or resveratrol (R). Body weight (**A**), relative kidney weight (kidney wet-weight /body weight ratio) (**B**), and serum AGEs levels (**C**) were measured. Data are presented as means ± SEM (n = 8 in each group). * *p* < 0.05, versus control group; # *p* < 0.05, versus DG group. DG: D-galactose, AG: aminoguanidine, R: resveratrol.

**Figure 2 metabolites-13-00655-f002:**
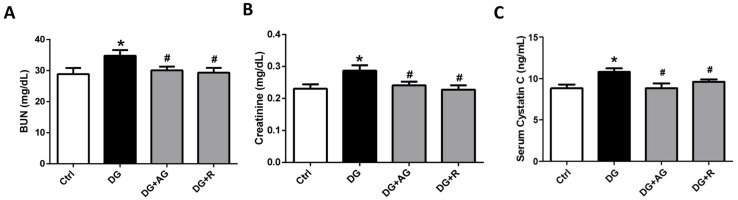
The renal function markers in D-galactose-induced aging mice. Mice were treated with D-galactose (DG) for 8 weeks in the presence or absence of either aminoguanidine (AG) or resveratrol (R). Blood samples were collected. Serum BUN (**A**), creatinine (**B**), and cystatin C (**C**) levels were measured. Data are presented as means ± SEM (n = 8 in each group). * *p* < 0.05, versus control group; # *p* < 0.05, versus DG group. DG: D-galactose, AG: aminoguanidine, R: resveratrol.

**Figure 3 metabolites-13-00655-f003:**
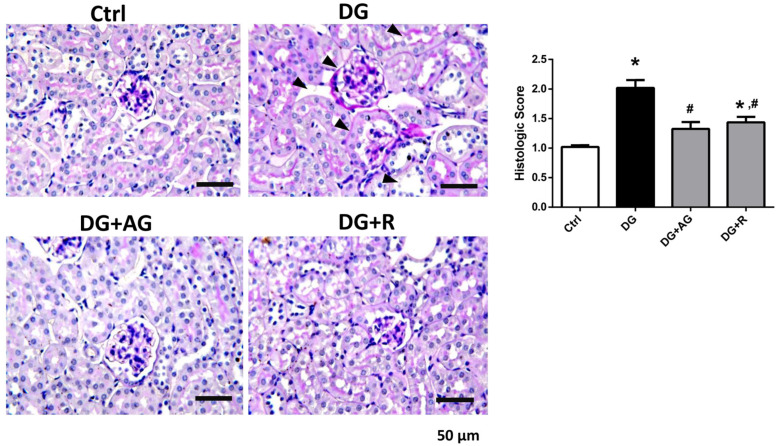
The histological examination in the kidneys of D-galactose-induced aging mice. Mice were treated with D-galactose (DG) for 8 weeks in the presence or absence of either aminoguanidine (AG) or resveratrol (R). Periodic acid–Schiff (PAS) staining was used to examine the histological changes in the kidneys. Arrow head indicates pathological changes. Histologic score was recorded. Data are presented as means ± SEM (n = 5 in each group). * *p* < 0.05, versus control group; # *p* < 0.05, versus DG group. Scale bar: 50 µm.

**Figure 4 metabolites-13-00655-f004:**
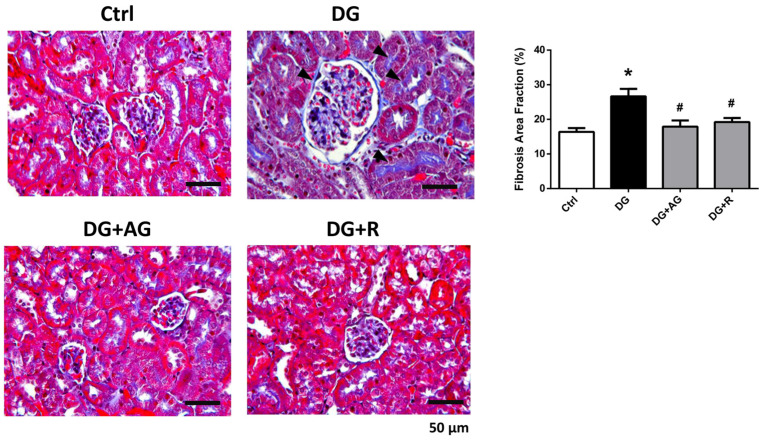
The fibrosis analysis in the kidneys of D-galactose-induced aging mice. Mice were treated with D-galactose (DG) for 8 weeks in the presence or absence of either aminoguanidine (AG) or resveratrol (R). Masson’s trichrome staining was used to analyze fibrosis in the kidneys. Arrow head indicates positive staining. The fibrosis area fraction was calculated by fibrotic area/whole tissue area. Data are presented as means ± SEM (n = 5 in each group). * *p* < 0.05, versus control group; # *p* < 0.05, versus DG group. Scale bar: 50 µm.

**Figure 5 metabolites-13-00655-f005:**
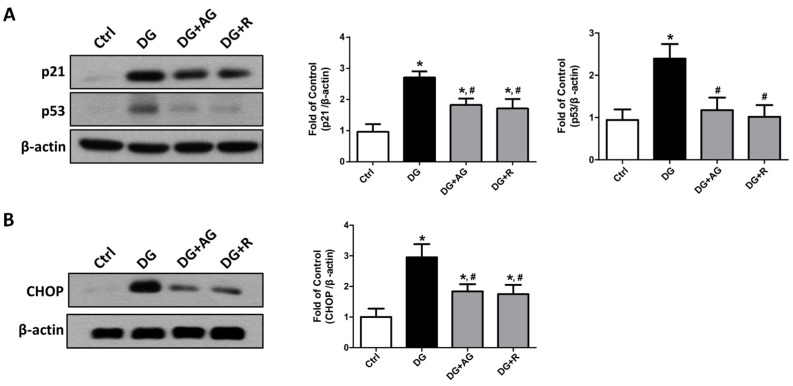
The changes in the cellular-senescence-related signaling molecules in the kidneys of D-galactose-induced aging mice. Mice were treated with D-galactose (DG) for 8 weeks in the presence or absence of either aminoguanidine (AG) or resveratrol (R). The levels of p21 and p53 (**A**) and CHOP (**B**) protein expression in the kidneys were shown. Protein levels were analyzed by Western blotting and quantified by densitometry and normalized by β-actin. Data are presented as means ± SEM (n = 4–6 in each group). * *p* < 0.05, versus control group; # *p* < 0.05, versus DG group. DG: D-galactose, AG: aminoguanidine, R: resveratrol.

**Figure 6 metabolites-13-00655-f006:**
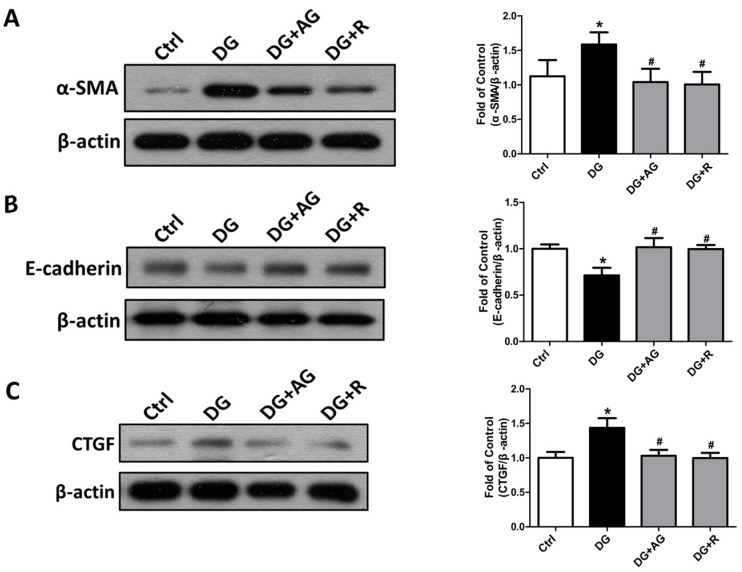
The changes in the epithelial-to-mesenchymal transition (EMT)-related signaling molecules in the kidneys of D-galactose-induced aging mice. Mice were treated with D-galactose (DG) for 8 weeks in the presence or absence of either aminoguanidine (AG) or resveratrol (R). The levels of α-SMA (**A**), E-cadherin (**B**), and CTGF (**C**) protein expression in the kidneys are shown. Protein levels were analyzed by Western blotting and quantified by densitometry and normalized by β-actin. Data are presented as means ± SEM (n = 4–6 in each group). * *p* < 0.05, versus control group; # *p* < 0.05, versus DG group. DG: D-galactose, AG: aminoguanidine, R: resveratrol.

**Figure 7 metabolites-13-00655-f007:**
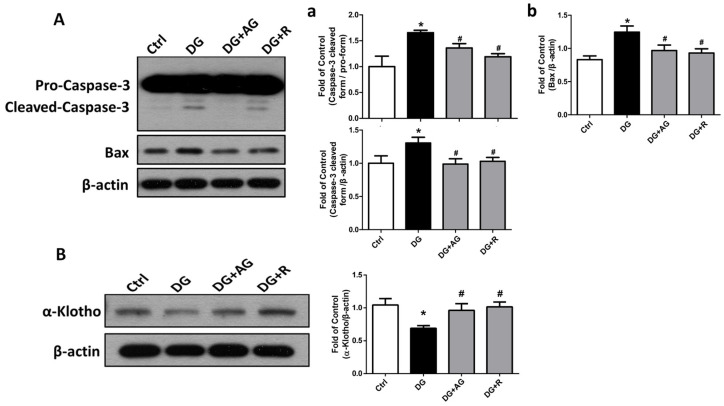
The changes in the apoptosis and survival-related signaling molecules in the kidneys of D-galactose-induced aging mice. Mice were treated with D-galactose (DG) for 8 weeks in the presence or absence of either aminoguanidine (AG) or resveratrol (R). The levels of caspase-3 (pro-caspase-3 and cleaved-caspase-3) and Bax (**A**) and α-Klotho (**B**) protein expression in the kidneys are shown. In (**a**), the ratio of caspase-3 cleaved-form and pro-form, and the ratio of caspase-3 cleaved form and β-actin ratio, are shown; in (**b**), the ratio of Bax and β-actin is shown. Protein levels were analyzed by Western blotting and quantified by densitometry and normalized by β-actin. Data are presented as means ± SEM (n = 4–6 in each group). * *p* < 0.05, versus control group; # *p* < 0.05, versus DG group. DG: D-galactose, AG: aminoguanidine, R: resveratrol.

## Data Availability

The data presented in this study are available from the corresponding author upon reasonable request. Data is not publicly available due to privacy or ethical restrictions.

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
