# Peer review of "Resveratrol Alleviates Advanced Glycation End-Products-Related Renal Dysfunction in D-Galactose-Induced Aging Mice"

_metabolites, 2023, doi:10.3390/metabo13050655_

Round 1

Reviewer 1 Report

In the paper “Resveratrol alleviates advanced glycation end-products-related renal dysfunction in D-galactose-induced aging mice” the authors explore the role of AGEs in renal function decline during aging and the protective effect of resveratrol on the process, in a D-galactose-induced aging mouse model. They found that serum levels of AGEs and renal dysfunction markers significantly increased after administration of D-galactose, and that the effects were significantly reversed by the AGEs inhibitor aminoguanidine and resveratrol treatment.

Major comments.

The effect of AGEs in renal dysfunction and AGEs in ageing are well known, as are the positive effects of resveratrol on ageing, renal dysfunction (refs. 18, 19) and AGEs production. The presence of AGEs in the murine ageing model of D-galactose has also already been demonstrated (ref. 16). Here are some examples in the last two years only 10.3390/ijms22158258, 10.1007/s11033-021-06550-3, 10.1016/j.bmcl.2021.127913, 10.3390/antiox12030584, 10.1002/biof.1531). If the authors want to proceed in this direction, they need to specify better what is new in their work and why this work is necessary.

Furthermore, as resveratrol also prevents oxidative stress, inflammation and renal fibrosis (that concomitantly act to induce ageing), it is difficult to claim that resveratrol treatment protects from AGEs-related kidney damage (rows 212).

Material and methods:

Section 2.1. Animals. The number of animals used in the experiments was not defined. Has a power analysis been carried out?

Section 2.3. Histological examination. How many sections were analysed per animal?

Section 2.4. Immunoblotting analyses. Was the lysate (and thus the biochemical analyses) performed on the entire kidney or only on sections?

Results and discussion.

Figure 3. It is difficult to appreciate the histological changes described by the authors. Higher magnification details and arrows may help the reader.

Figures 4-6. Signal saturation is often evident in representative western blot images (actin in all blots). Less saturated blots should be presented. Also, because there is variability between animals, more samples (at least three animals per group) should be present in the same blot. Whole blots must be provided in the supplementary material.

In Figure 6A, the signal of the pro-caspase-3 band is saturated, two exposures (one for pro-caspase and one for clived-caspase) should be shown. In the blot, there is no signal difference in cleaved-caspase between the DG and the DG+rev samples, as described in the text and as quantified in the histogram, please provide a more representative image. Apparently the two histograms in fig 6Aa show quantification of the same band (cleaved caspase 3), please reconsider.

Minors

Figure 3. The D-galactose bar should be black, according to the other figures

Row 58. “D-galactose-induced aging animal model was widely applied in anti-aging research”. There are no bibliographic references at this point.

A revision of the English language (style and sentence construction) is needed, especially in the introduction, where it is sometimes difficult to understand the author's point of view

Author Response

Reviewer #1

In the paper “Resveratrol alleviates advanced glycation end-products-related renal dysfunction in D-galactose-induced aging mice” the authors explore the role of AGEs in renal function decline during aging and the protective effect of resveratrol on the process, in a D-galactose-induced aging mouse model. They found that serum levels of AGEs and renal dysfunction markers significantly increased after administration of D-galactose, and that the effects were significantly reversed by the AGEs inhibitor aminoguanidine and resveratrol treatment.

(1). Major comments.

The effect of AGEs in renal dysfunction and AGEs in ageing are well known, as are the positive effects of resveratrol on ageing, renal dysfunction (refs. 18, 19) and AGEs production. The presence of AGEs in the murine ageing model of D-galactose has also already been demonstrated (ref. 16). Here are some examples in the last two years only 10.3390/ijms22158258, 10.1007/s11033-021-06550-3, 10.1016/j.bmcl.2021.127913, 10.3390/antiox12030584, 10.1002/biof.1531). If the authors want to proceed in this direction, they need to specify better what is new in their work and why this work is necessary.

Response: We appreciate the comment by reviewer. We have tried our best to revise this manuscript according to the suggestion of reviewer. We have cited these previous studies and added a discussion in this revised manuscript according to the suggestion of reviewer.

“There were several review articles for the effects of AGEs or resveratrol on aging. AGEs induce pathophysiological changes during the aging process [37, 38]. AGEs are known the aging products, and participates in the onset and progress of chronic kidney disease [39]. Resveratrol has been suggested to have the potential to suppress the AGEs-related complications [40]. Resveratrol was found to possess the anti-aging effects in the kidneys by improving pathologies and molecular signals, including AMP-activated protein kinase (AMPK), nuclear factor-κB, and sirtuin-1 [41]. Vlassara et al. (1994) observed the pathogenic role of AGEs in the induction of glomerular sclerosis and albuminuria in healthy rats administered intravenously with AGE-modified rat albumin, and the effects could be ameliorated by aminoguanidine treatment [42]. Moreover, Abharzanjani and Hemmati (2021) found that resveratrol (10 and 15 mg/kg, i.p. for 3 weeks) could prevent the kidney complications and aging in streptozotocin-induced diabetic rats [43]. Resveratrol (5 mg/kg, oral for 4 weeks) has been found to decrease elevated serum AGEs levels and alleviate renal dysfunction in streptozotocin‒nicotinamide-induced diabetic rats [44]. Resveratrol (10 mg/kg, oral for 4 weeks) has also been suggested to have therapeutic potential in malignantly hypertensive rats against kidney injury [45]. However, the mechanisms of aminoguanidine or resveratrol on aging kidney remains to be clarified.

Several studies with experimental designs using D-galactose-induced aging mouse model. Chu et al. (2021) showed that the expression levels of the klotho gene and protein in the kidneys of D-galactose-injected (1 g/kg, intraperitoneal injection) mice were significantly lower than those of the control group, and the klotho expression was enhanced by a resveratrol (12.5, 25, and 50 mg/kg, oral administration for 4 weeks) treatment [46]. Fan et al. (2016) observed that increased AGEs levels in the kidneys of D-galactose-injected (120 mg/kg, subcutaneous injection) mice, as compared to a control group [15]. Li et al. (2021) found that resveratrol (10-40 mg/kg, oral administration for 8 weeks) significantly downregulated the abnormal increases in BUN and creatinine serum levels, caused by D-galactose (100 mg/kg, intraperitoneal injection) treatment, and effectively improved pathological damage [47]. In the present study, we observed increased serum levels of AGEs and the markers of renal function and renal histopathological damage and fibrosis, as well as the inductions of cellular senescence, fibrosis, and apoptosis in the kidneys of D-galactose-induced aging mice (1 g/kg, subcutaneous injection), and these changes could be significantly reversed by treatment with AGEs inhibitor aminoguanidine (100 mg/kg, oral) or resveratrol (40 mg/kg, oral) for 8 weeks. Our study had a similar experimental design using D-galactose aging mouse model to the aforementioned studies, but we also had novel findings for simultaneously comparing the effects of aminoguanidine and resveratrol, including the improvement of pathological characterizations determined by the histological score and collagen deposition (Masson’s trichrome staining) and the protective mechanisms of aminoguanidine or resveratrol against cellular senescence, fibro-sis-related EMT, and apoptosis in aging kidneys.”

(2). Furthermore, as resveratrol also prevents oxidative stress, inflammation and renal fibrosis (that concomitantly act to induce ageing), it is difficult to claim that resveratrol treatment protects from AGEs-related kidney damage (rows 212).

Response: We appreciate the comment by reviewer. We have revised this sentence in this revised manuscript according to the suggestion of reviewer.

“These findings suggest that AGEs play an important role in renal dysfunction during the aging process, and resveratrol treatment alleviates AGEs-related renal dysfunction through the improvement of renal cellular senescence, apoptosis, and fibrosis-related EMT in D-galactose-induced aging in mice.” (rows 355-358)

(3). Material and methods:

Section 2.1. Animals. The number of animals used in the experiments was not defined. Has a power analysis been carried out? Section 2.3. Histological examination. How many sections were analysed per animal? Section 2.4. Immunoblotting analyses. Was the lysate (and thus the biochemical analyses) performed on the entire kidney or only on sections?

Response: We appreciate the comment by reviewer. We have revised the Materials and methods section in this revised manuscript according to the suggestion of reviewer.

“The eight mice per group was used.

A power analysis has been carried out in this study. The post hoc power calculation was performed using G*Power software (latest ver. 3.1.9.7; Heinrich-Heine-Universität Düsseldorf, Düsseldorf, Germany). The results showed that with a total sample size of 32 (n = 8 per group), we had 84% statistical power to detect a significant effect for creatinine value at a 5% significance level, with a total sample size of 32 (n = 8 per group).

For histological examination, fifteen fields per section per animal were assessed.

For immunoblotting, kidney cortical lysates were prepared. Kidney sections (10-15 mg) were lysed in 200 μL radioimmunoprecipitation assay (RIPA) lysis buffer.”

(4). Results and discussion.

Figure 3. It is difficult to appreciate the histological changes described by the authors. Higher magnification details and arrows may help the reader.

Figures 4-6. Signal saturation is often evident in representative western blot images (actin in all blots). Less saturated blots should be presented. Also, because there is variability between animals, more samples (at least three animals per group) should be present in the same blot. Whole blots must be provided in the supplementary material.

In Figure 6A, the signal of the pro-caspase-3 band is saturated, two exposures (one for pro-caspase and one for clived-caspase) should be shown. In the blot, there is no signal difference in cleaved-caspase between the DG and the DG+rev samples, as described in the text and as quantified in the histogram, please provide a more representative image. Apparently the two histograms in fig 6Aa show quantification of the same band (cleaved caspase 3), please reconsider.

Response: We appreciate the comment by reviewer. We have revised the presentation of figures in this revised manuscript according to the suggestion of reviewer.

“The original Figure 3 was divided into Figure 3 (histological examination) and Figure 4 (Masson’s trichrome stain), and the higher magnification was shown.

The relatively high saturation of these blot films may be due to the strong antibody signals. These blot films are not modified by adjusting contrast or brightness, and are the original appearance.

We agree with the comment by the reviewer that there is variability between animals, more samples (at least three animals per group) should be present. The original data (at least three animals per group) for whole blots are presented in a supplemental file according to the suggestion of reviewer.

Moreover, the Figure 6Aa has been revised that cleaved caspase 3/beta-actin ratio is deleted, and cleaved caspase 3/pro-caspase 3 is kept according to the suggestion of reviewer.”

(5). Minors

Figure 3. The D-galactose bar should be black, according to the other figures

Row 58. “D-galactose-induced aging animal model was widely applied in anti-aging research”. There are no bibliographic references at this point.

Response: We appreciate the comment by reviewer. We have revised the Figure 3 and added the references for sentence in original Row 58 (Row 61 in this revised manuscript) according to the suggestion of reviewer.

Reviewer 2 Report

The submitted study describes the influences of aging induced by long-term D-galactose treatment on AGE levels, renal function and expression on signaling molecules, and the alleviation effects of resveratrol on them. The results of the submitted study, that resveratrol reverses various adverse effects induced by D-galactose treatment, provide useful information for utilization of resveratrol as an anti-aging substance. Therefore, the submitted study would be acceptable to Metabolites. However, there are some points to be considered before the publication. Please consider and revise the manuscript according to the comments listed below.

1.     The dose of anti-aging substances administered to the mice was 100 mg/kg mg for aminoguanidine while 40 mg/kg for resveratrol. Please add an explanation of how these doses were determined.  Also, did the authors confirm that the effect of resveratrol was enhanced with increasing dosage?

2. Although not cited in the present manuscript, several studies with experimental designs similar to the submitted study have been reported as follows. Y. Fan et al. reported that AGEs in the kidney of D-galactose model mice were increased compared with the normal group (Pharmaceutical Biology, 2016, 54(9), 1815-1821 [https://doi.org/10.3109/13880209.2015.1129543]). S.-H. Chu et al. reported the expression of klotho gene and protein in kidney tissues of D-galactose model mice was significantly lower than those of normal group mice, and the klotho expression was enhanced obviously with the resveratrol treatment (Bioorganic & Medicinal Chemistry Letters, 2021, 40, 127913 [https://doi.org/10.1016/j.bmcl.2021.127913]). L. Li et al. reported that resveratrol significantly downregulated abnormal increases in serum levels of BUN and creatinine caused by long-term D-galactose treatment, which effectively improved pathological damage (Food and Function, 2021, 12(18), 8274-8287 [https://doi.org/10.1039/D1FO00538C]). Please cite these studies and consider adding a discussion of the similarities and differences between the results of the submitted study and those of previous studies.

3. Recently, J.-Y. Sha et al. reported the ameliorative effects of maltol on D-galactose-induced kidney aging and injury (Phytotherapy Research, 2021, 35(8), 4411-4424 [https://doi.org/10.1002/ptr.7142]). Similar to resveratrol described in the submitted study, maltol was reported to reverse the increase in the protein expressions of p53, p21, Bax and caspase-3 induces by D-galactose treatment. How effective is resveratrol compared to previously reported other anti-aging substances such as maltol and aminoguanidine?

Author Response

Reviewer #1

The submitted study describes the influences of aging induced by long-term D-galactose treatment on AGE levels, renal function and expression on signaling molecules, and the alleviation effects of resveratrol on them. The results of the submitted study, that resveratrol reverses various adverse effects induced by D-galactose treatment, provide useful information for utilization of resveratrol as an anti-aging substance. Therefore, the submitted study would be acceptable to Metabolites. However, there are some points to be considered before the publication. Please consider and revise the manuscript according to the comments listed below.

  1. The dose of anti-aging substances administered to the mice was 100 mg/kg mg for aminoguanidine while 40 mg/kg for resveratrol. Please add an explanation of how these doses were determined.  Also, did the authors confirm that the effect of resveratrol was enhanced with increasing dosage?

Response: We appreciate the comment by reviewer. We have added an explanation of how these doses were determined in this revised manuscript according to the suggestion of reviewer.

In our preliminary study, we have used the 60 mg/kg dose of resveratrol, but the effect was no better.

“The dosage selection for aminoguanidine and resveratrol was according to the previous studies [21, 22] and our preliminary experiments.”

  1. Although not cited in the present manuscript, several studies with experimental designs similar to the submitted study have been reported as follows. Y. Fan et al. reported that AGEs in the kidney of D-galactose model mice were increased compared with the normal group (Pharmaceutical Biology, 2016, 54(9), 1815-1821 [https://doi.org/10.3109/13880209.2015.1129543]). S.-H. Chu et al. reported the expression of klotho gene and protein in kidney tissues of D-galactose model mice was significantly lower than those of normal group mice, and the klotho expression was enhanced obviously with the resveratrol treatment (Bioorganic & Medicinal Chemistry Letters, 2021, 40, 127913 [https://doi.org/10.1016/j.bmcl.2021.127913]). L. Li et al. reported that resveratrol significantly downregulated abnormal increases in serum levels of BUN and creatinine caused by long-term D-galactose treatment, which effectively improved pathological damage (Food and Function, 2021, 12(18), 8274-8287 [https://doi.org/10.1039/D1FO00538C]). Please cite these studies and consider adding a discussion of the similarities and differences between the results of the submitted study and those of previous studies.

Response: We appreciate the comment by reviewer. We have cited these studies and added a discussion in this revised manuscript according to the suggestion of reviewer.

“Several studies with experimental designs were similar to this study. Fan et al. (2016) observed that increased AGEs levels in the kidneys of D-galactose-injected (120 mg/kg, subcutaneous injection) mice, as compared to a control group [15]. Chu et al. (2021) showed that the expression levels of the klotho gene and protein in the kidneys of D-galactose-injected (1 g/kg, intraperitoneal injection) mice were significantly lower than those of the control group, and the klotho expression was enhanced by a resveratrol (12.5, 25, and 50 mg/kg, oral administration for 4 weeks) treatment [42]. Li et al. (2021) found that resveratrol (10-40 mg/kg, oral administration for 8 weeks) significantly downregulated the abnormal increases in BUN and creatinine serum levels, caused by D-galactose (100 mg/kg, intraperitoneal injection) treatment, and effectively improved pathological damage [43]. In the present study, we observed increased serum levels of AGEs and the markers of renal function and renal histopathological damage, as well as the inductions of cellular senescence, fibrosis, and apoptosis in the kidneys of D-galactose-induced aging mice (1 g/kg, subcutaneous injection), and these changes could be significantly reversed by treatment with aminoguanidine or resveratrol (40 mg/kg, oral administration for 8 weeks). Our study had a similar experimental design to the aforementioned studies, but we also had novel findings, including the pathological characterizations of the D-galactose aging mouse model and the protective mechanism of resveratrol against renal damage in aging kidneys.”

  1. Recently, J.-Y. Sha et al. reported the ameliorative effects of maltol on D-galactose-induced kidney aging and injury (Phytotherapy Research, 2021, 35(8), 4411-4424 [https://doi.org/10.1002/ptr.7142]). Similar to resveratrol described in the submitted study, maltol was reported to reverse the increase in the protein expressions of p53, p21, Bax and caspase-3 induces by D-galactose treatment. How effective is resveratrol compared to previously reported other anti-aging substances such as maltol and aminoguanidine?

Response: We appreciate the comment by reviewer. We have cited this study and added a discussion in this revised manuscript according to the suggestion of reviewer.

“Sha et al. (2021) found that maltol reversed the increase in the protein expression of p53, p21, Bax, and caspase-3, induced by D-galactose treatment, and suggested that a phosphatidylinositol-3-kinase (PI3K)/protein kinase B (Akt) signaling pathway contributed to the protective effect of maltol against cellular senescence and injury of the liver and kidney in D-galactose-induced aging mice [44]. In the present study, we also found that resveratrol could effectively reverse the increased cellular senescence and apoptosis in the kidneys of D-galactose-induced aging mice. The PI3K/Akt signaling pathway may be involved in the protective effect of resveratrol against kidney aging and injury, which should be studied further in the future.”

Reviewer 3 Report

The manuscript describes the ability of resveratrol to prevent the production of AGEs following D-galactose-induced aging in mice, and the prevention of renal dysfunction.

Extensive English styling is required.

Line 36 has a syntactic problem.

Line 40: Instead of "was started to decrease" you can write: "starts to decrease".

Line 48: Instead of "would occur" you can write "occur"

Lines 52-53: Instead of "highly paid attention" you can write "have attracted attention"; instead of "lots of" you can write "several" and find another expression for "are investigated" – maybe "several mechanisms have been suggested".

Line 58: Correct to: "is widely applied" (should be in presence tense).

Line 59: Correct to: "Several natural extracts and agents such as…., have been shown to have anti-aging activities using this animal model".

Line 61: remove "the" before "anti-oxidative".

The next sentences should be split into separate sentences, and styled.

Line 63: Is it a decrease of AGEs that have already been produced, or is it a prevention f its production. Please describe.

Line 72: Correct to: the potential".

The last sentence of Introduction needs to be rewritten.

Line 79: Correct to: "10-week-old C57BL/6 male mice"

Line 89: Instead of "provided" I would write "administrated" or "injected subcutaneously".

What were the controls of the experiments?

Did you have a control group receiving caboxymethylcellulose?

Line 101: The competitive ELISA assay should be described in detail and the catalog number of the ELISA kits should be provided.

Line 104: Please state the composition of the buffered formalin.

Line 105: Describe the staining method.

Line 114: How much kidney tissue was taken for lysis, and how much RIPA was used for each sample? Please state these parameters.

Line 123: The concentration of Tris and glycine should be stated.

The catalog number of the different antibodies should be stated.

Line 134: Still using film or an imager?

Line 145: Use another word than "mimetic"

What are the adverse effects of resveratrol?

Lines 146-147 should be rewritten.

It is better to have one Result section and then the Discussion section. In the current state – the text is not organized and clear.

Bw should be defined first time used.

Line 159: "the four groups"

The whole Result section should be edited by a scientific English editor.

Line 162: "Both" agents together? Or each agent separately? Thus, the sentences should be rewritten. The same goes for other places where you have used the word "both".

The letter size in the Figures should be larger and readable.

Figure 1: Should be in the X-axis AG and not Ag. The same goes for the other figures.

Title of Figure 1 is not accurate. Remove "changes of". The same for Figure 2 and 3.

Is the kidney weight wet weight? If so, state it.

The authors didn't show the full WB images.

Needs extensive English editing.

Author Response

Reviewer #2

The manuscript describes the ability of resveratrol to prevent the production of AGEs following D-galactose-induced aging in mice, and the prevention of renal dysfunction.

  • Extensive English styling is required.

Line 36 has a syntactic problem.

Line 40: Instead of "was started to decrease" you can write: "starts to decrease".

Line 48: Instead of "would occur" you can write "occur"

Lines 52-53: Instead of "highly paid attention" you can write "have attracted attention"; instead of "lots of" you can write "several" and find another expression for "are investigated" – maybe "several mechanisms have been suggested".

Line 58: Correct to: "is widely applied" (should be in presence tense).

Line 59: Correct to: "Several natural extracts and agents such as…., have been shown to have anti-aging activities using this animal model".

Line 61: remove "the" before "anti-oxidative".

The next sentences should be split into separate sentences, and styled.

Line 63: Is it a decrease of AGEs that have already been produced, or is it a prevention  its production. Please describe.

Line 72: Correct to: the potential".

The last sentence of Introduction needs to be rewritten.

Response: We appreciate the reviewer's comments. We have revised this manuscript with extensive English styling according to the suggestion of reviewer. Moreover, this revised manuscript has been received an English editing service by MDPI (English Editing Invoice ID: english-64972).

  • Line 79: Correct to: "10-week-old C57BL/6 male mice"

Line 89: Instead of "provided" I would write "administrated" or "injected subcutaneously".

What were the controls of the experiments?

Did you have a control group receiving caboxymethylcellulose?

Line 101: The competitive ELISA assay should be described in detail and the catalog number of the ELISA kits should be provided.

Line 104: Please state the composition of the buffered formalin.

Line 105: Describe the staining method.

Line 114: How much kidney tissue was taken for lysis, and how much RIPA was used for each sample? Please state these parameters.

Line 123: The concentration of Tris and glycine should be stated.

The catalog number of the different antibodies should be stated.

Line 134: Still using film or an imager?

Line 145: Use another word than "mimetic"

Response: We appreciate the reviewer's comments. We have revised this manuscript according to the suggestions of reviewer.

(A). We have revised this manuscript with extensive English styling according to the suggestion of reviewer. Moreover, this revised manuscript has been received an English editing service by MDPI (English Editing Invoice ID: english-64972).

(B). We have also added the description for the controls of experiments in the Methods section. The control group received caboxymethylcellulose solution.

  • What are the adverse effects of resveratrol?

Lines 146-147 should be rewritten.

It is better to have one Result section and then the Discussion section. In the current state – the text is not organized and clear.

Bw should be defined first time used.

Line 159: "the four groups"

The whole Result section should be edited by a scientific English editor.

Line 162: "Both" agents together? Or each agent separately? Thus, the sentences should be rewritten. The same goes for other places where you have used the word "both".

The letter size in the Figures should be larger and readable.

Figure 1: Should be in the X-axis AG and not Ag. The same goes for the other figures.

Title of Figure 1 is not accurate. Remove "changes of". The same for Figure 2 and 3.

Is the kidney weight wet weight? If so, state it.

The authors didn't show the full WB images.

Response: We appreciate the comment by reviewer. We have revised this manuscript according to the suggestions of reviewer.

(A). We have re-organized the structure of this manuscript, which has one Result section and then the Discussion section.

(B). We have added a discussion for the adverse effects of resveratrol at the end of Discussion section.

“Resveratrol has been suggested to be a double-edged sword for its benefit effects on health [45]. It exhibited several bioactive effects, such as anti-inflammatory, anti-cancer, cardiovasculoprotective, and neuroprotective. However, it has some concerns for pharmaceutical industry, such as poor solubility and bioavailability and adverse effects [45]. The side effects of mild to moderate gastrointestinal symptoms by resveratrol have been observed at doses of 2.5 and 5 g per day for 29 days in healthy volunteers [46]. The dosage and route of administration involved in the safety of resveratrol still need to be considered and explored.”

(C). We have revised this manuscript with extensive English styling according to the suggestion of reviewer. Moreover, this revised manuscript has been edited by an English editing service of MDPI (English Editing Invoice ID: english-64972).

(D). The full WB images were shown in a supplemental data file.

Round 2

Reviewer 1 Report

No further comments

Author Response

Reviewer 1:

No further comments.

Response: We appreciate the reviewer for their positive affirmation of our work. There are no further comments by reviewer.

Reviewer 2 Report

The authors have made appropriate additions to the discussion in accordance with previous comments and the manuscript would have been improved. Therefore, the revised manuscript would be acceptable for publication in its present form.

Author Response

Reviewer 2

The authors have made appropriate additions to the discussion in accordance with previous comments and the manuscript would have been improved. Therefore, the revised manuscript would be acceptable for publication in its present form.

Response: We appreciate the reviewer for their positive affirmation of our work. There are no further comments by reviewer.

Reviewer 3 Report

Comments to revised metabolites -2361321.

The manuscript has been improved.

The are some issues that need to be addressed before the manuscript can be accepted for publication:

Line 55: Correct to: "to cause"

Line 275: correct to "shown"

Line 292: Correct to: "AGEs can induce"

Lines 314 and 317 and other places: Capital K in " α-Klotho"

Section starting at line 318 and the further paragraphs need English editing.

Line 318: e.g., Correct to: " There are several review articles discussing the effects of AGEs or resveratrol on aging."

Line 319: e.g., Correct to: " AGEs are known aging products"

Line 323: What do you mean with "improving"? Is it an increase or decrease? Please be more specific.

Please let an English editor go over the discussion (too many syntactic problems).

Also, the discussion is more like a mentioning of data in the literature, but should integrate the data presented in the manuscript and emphasize what are the novel data the authors have revealed.

The new sections need English editing.

Author Response

Response to Reviewer 3

Comments to revised metabolites -2361321.

The manuscript has been improved.

The are some issues that need to be addressed before the manuscript can be accepted for publication:

(1). Line 55: Correct to: "to cause"

Line 275: correct to "shown"

Line 292: Correct to: "AGEs can induce"

Lines 314 and 317 and other places: Capital K in " α-Klotho"

Section starting at line 318 and the further paragraphs need English editing.

Line 318: e.g., Correct to: " There are several review articles discussing the effects of AGEs or resveratrol on aging."

Line 319: e.g., Correct to: " AGEs are known aging products"

Line 323: What do you mean with "improving"? Is it an increase or decrease? Please be more specific.

Please let an English editor go over the discussion (too many syntactic problems).

Response: We appreciate the comment by reviewer. We have carefully revised the language of this manuscript especially the Discussion section according to the suggestion of reviewer.

We also improve English writing in this revised manuscript through a professional English editing service [Multidisciplinary Digital Publishing Institute (MDPI), Basel, Switzerland; https://www.mdpi.com/authors/english) (English Editing Invoice ID: english-64972)].

(2). Also, the discussion is more like a mentioning of data in the literature, but should integrate the data presented in the manuscript and emphasize what are the novel data the authors have revealed.

Response: We appreciate the comment by reviewer. We have revised the Discussion in this revised manuscript according to the suggestion of reviewer.

We integrated the data presented in the manuscript and emphasize what are the novel data the authors have revealed, such as follows:

“Compared with previous studies, this study has different approaches and novel find-ings. We investigated the protective effect and mechanism of resveratrol (40 mg/kg, oral for 8 weeks) against kidney aging and injury using a D-galactose-induced aging mouse model. We also compared the effects of resveratrol and AGEs inhibitor amino-guanidine to clarify the role of AGEs in it. We found that resveratrol could alleviate AGEs-related renal dysfunction by the decreases in renal cellular senescence, apoptosis, and fibrosis-related EMT in aging mice.” (lines 335-341)

Round 3

Reviewer 3 Report

The manuscript has been significantly improved and is now suitable for publication.